# Approaches to stable isotope tracing and in vivo metabolomics in the cancer clinic

Brandon Faubert [1]✉ & Alpaslan Tasdogan [2]✉

Metabolic reprogramming in cancer cells is complex. Cancer cells rewire nutrient uptake and enzyme activity to support malignant properties. These metabolic phenotypes are the combination of intrinsic factors, including mutation status or tissue of origin, and extrinsic factors, such as alterations in blood supply or the contributions of multiple cell types within the tumor microenvironment. This complexity can be challenging to reproduce in vitro and has prompted investigators to study tumor metabolism in vivo, including directly in patients. By administering nutrients labeled with a stable isotope to patients, investigators can track labeled atoms through biochemical reactions, providing a detailed assessment of tumor metabolism. In this perspective, we discuss stable isotope tracing in cancer patients from the perspective of protocol initiation, the workflows and pitfalls in establishing these procedures, and offer insights into analyzing tumor metabolism from patient samples.

Keywords Intraoperative Patient Infusions; Stable Isotope Tracing; Cancer Metabolism
Subject Categories Metabolism; Methods & Resources

See also: Metabolism Methods Commentary Series

Metabolic processes involve the continuous synthesis, breakdown, and conversion of chemical compounds that sustain biological function. In cancer, deregulated cellular metabolism is a hallmark of malignancy, as these cells rewire metabolic programs and nutrient acquisition strategies to support sustained growth and proliferation. Recent efforts have focused on interrogating tumor metabolism in vivo by adding stable isotope tracers, e.g., nutrients labeled with a heavy (non-radioactive) element. These labeled nutrients are metabolized similarly to non-labeled counterparts, allowing investigators to assess nutrient preferences, relative enzymatic activity, and the contribution of metabolites through various pathways. Stable isotopes have been used to measure metabolism in people for decades (Kim et al, 2016), but only recently have been applied to patients with cancer (Fan et al, 2009). These works have provided unique insight into nutrient utilization by tumors, revealing heterogeneous metabolic phenotypes within and between tumors. While efforts to apply these methods in patients continue to grow, establishing any tracing study in human subjects brings unique challenges and using these protocols in the cancer clinic can add additional complexity.

These studies often begin with a straightforward question or hypothesis. What nutrients do tumors use in their native environment? How do specific oncogenes or organ environments influence metabolism in human tumors? Does the metabolism of a primary tumor correlate with clinical features like drug resistance? Yet, while many protocols detail how to process tissue samples and analyze stable isotope tracing data (Faubert et al, 2021; Kim et al, 2016), few published accounts detail the practical applications of establishing infusion protocols in cancer patients. While navigating the approval process and identifying the requisite safeguards can be achieved with the help of Institutional Review Boards, the specifics of applying stable isotope infusions in patients with cancer, such as sampling, operational procedures, timing, and patient selection, can be vague. Here, we aim to guide the reader through some pitfalls and considerations regarding these protocols, as summarized in Fig. 1.

## Study design: safety and regulatory considerations

All clinical infusion studies begin (or end) with the IRB. Their fundamental objective is safeguarding the research participants' rights and well-being and upholding ethical standards. Further, the IRB ensures compliance with regulatory guidelines, which will vary in each country. For instance, in Germany, the Federal Institute for Drugs and Medical Devices (BfArM) and the local IRB will judge the safety of every product, including nutrients labeled with stable isotopes. In the United States, the American Food and Drug Administration (FDA) has generally approved using nutrients labeled with stable isotopes in patients, provided several conditions are met. Some conditions include (a) that the quality of the tracer meets relevant clinical standards, (b) that the research is intended to obtain basic information on the metabolism of the substrate, (c) that the study is not intended for immediate benefit (e.g., therapeutic or diagnostic applications) to the subject and (d) that the dose of the labeled metabolite is not known to cause clinically detectable side effects. Provided these prerequisites are satisfied, the FDA generally does not object to using stable isotope tracers in patients.

Various stable isotopes (e.g., $^{13}C$, $^{15}N$, $^{2}H$) have been used in humans and are generally considered to have identical chemical properties and behaviors to the common isotopes. When used as a nutrient tracer, the effects on enzyme kinetics are limited for $^{13}C$ and $^{15}N$, even at concentrations that exceed those typically employed in physiological studies. $^{2}H$ has been the subject of further debate, with some studies indicating high abundances of $^{2}H$-labeled

[1]Department of Medicine, Section of Hematology/Oncology, University of Chicago, Chicago, IL, USA. [2]Department of Dermatology, University Hospital Essen & German Cancer Consortium (DKTK), Partner Site Essen, Germany. ✉E-mail: bfaubert@bsd.uchicago.edu; alpaslan.tasdogan@uk-essen.de
https://doi.org/10.1038/s44318-025-00450-z | Published online: 12 May 2025

**Glossary**

| | |
|---|---|
| Bolus | A single dose of a $^{13}$C-labeled nutrient is often administered at a higher dose to increase the enrichment of the nutrient pool. |
| Cell isolation | Methods of disaggregating a tumor or tissue into a cell suspension and, subsequent isolation of single cell populations. |
| Isotopic steady state | Adding a $^{13}$C-labeled substrate will enrich down-stream metabolites as the tracer is metabolized in a |

| | |
|---|---|
| | time-dependent fashion. "Steady-state" is achieved when the $^{13}$C enrichment of a given metabolite is stable over time. |
| Spatial metabolomics | A collection of techniques/instruments that measure metabolite levels in dimensional space (e.g., across the x and y axis of a tissue slice). |
| Stable isotope tracer | A molecule (usually a nutrient) that has one or more stable isotopes. |

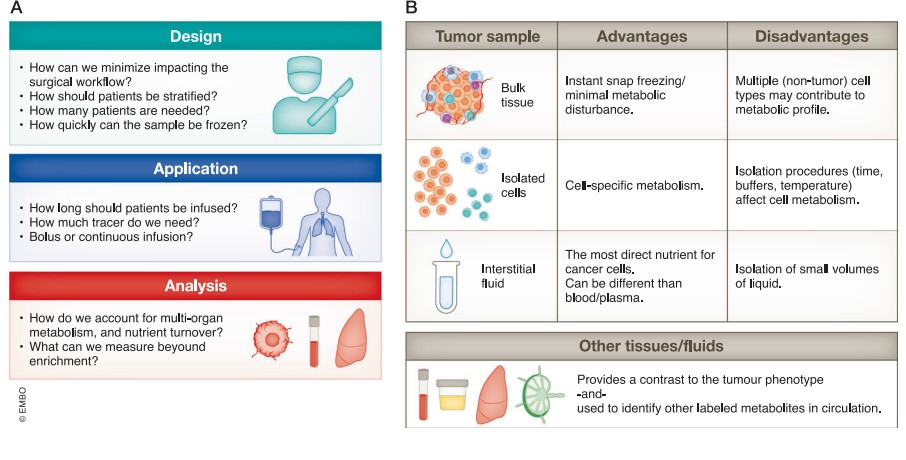

**Figure 1. Considerations and advantages of stable isotope tracing in patients with cancer.**

(A) General considerations for performing stable isotope tracing in the clinical setting, including study design, applying specific tracer conditions, and considerations for analysis. (B) Strategies for tissue sample analysis and potential drawbacks for each approach. Other bodily fluids, such as blood and urine can be collected for comparative analysis. Similarly, depending on the clinical procedure, adjacent non-malignant tissue from the same organ can be collected for analysis.

molecules can detrimentally impact enzymatic activity (Thomson, 1960). In humans, very high doses of $^2H_2O$ have been associated with transient episodes of vertigo. However, there is a lack of evidence for other adverse effects at low doses, as other $^2$H-labeled metabolites have been successfully administered with few reported side effects. Thus, primary safety considerations for tracing studies in patients with cancer include (a) potential effects from nutrient dosage (e.g., avoiding hyperglycemia and insulin response when infusing [$^{13}$C] glucose), (b) that the infusion protocols do not significantly alter standard operating procedures (e.g., extending the duration in which the patient is under anesthesia), and (c) the safety considerations of the investigational product itself. At a minimum, microbiological and pyrogen-tested (MPT) material is required for human infusions, while some institutions require tracers to be clinical trial material (CTM) grade. Accounting for various safety considerations is generally manageable, as infusion protocols can be designed with minimal interruptions to the surgical workflow. However, a significant hurdle to investigators is the economic challenge of these studies due to the costs of MPT or CTM-grade $^{13}$C-labeled tracers.

## Tracer administration and patient selection

Labeled nutrients are often administered as a single bolus or a primed-continuous infusion. Either method can be used with minimal disruption of standard surgical procedures, so the research question and type of desired data determine the choice in methodology. Providing a pre-operative bolus of a labeled nutrient offers several advantages, including ease of use and a minimal amount of labeled nutrient needed. A drawback of this approach is that a bolus may not provide adequate signal in metabolites or pathways that take longer to develop. A primed-continuous infusion enables maximum representation of the potential metabolite labeling products, providing detailed insight into tumor metabolism. A priming dose is used to rapidly elevate the contraction of the tracer, which reduces the time needed to reach isotopic steady-state, wherein the $^{13}$C enrichment is stable over time. Reaching steady-state depends upon several factors, including the size and turnover rate of the endogenous metabolite pool, as well as the sampling site. In other words, steady-state occurs at different times in different tissues. For instance, in patients who have fasted overnight before surgery, $^{13}$C glucose infusions reach a plateau relatively rapidly, whereas the TCA cycle of lung tumors can take two or more hours (Faubert et al, 2017). A notable drawback of this approach is that continuous infusion typically requires greater amounts of tracer, as the prepared infusion mix needs to last the duration of the procedure, plus extra material in case of unexpected delays.

After deciding the most effective means of administering the stable isotope, which patients should be infused? How many patients are needed? Patient stratification can be essential in accounting for potential variability in tumor metabolism, as numerous patient-level factors (e.g., age, sex, BMI, co-morbidities) can potentially influence tumor metabolism. Yet, how many patients are needed to evaluate significant differences in tumor metabolism can be difficult to assess. Based on the studies to date, it has become apparent (at least for glucose metabolism) that tumors have heterogeneous metabolism, even when [$^{13}$C]glucose dose and infusion parameters are constant between studies. For instance, primary kidney tumors consistently have low [$^{13}$C] glucose-derived enrichment in the TCA cycle. In contrast, lung, brain, and other tumor types have shown increased enrichment, which can be highly variable within each tumor type (Courtney et al, 2018; Fan et al, 2009; Ghergurovich et al, 2021). With this variability in mind, it may be beneficial to approach a new infusion protocol as a feasibility trial (and a small number of initial patients recruited) rather than testing for specific differences between tumor groups. Maximizing the types of samples

acquired (tumor tissue, adjacent, non-malignant tissues, blood, urine) and detailed downstream analyses (expanded upon below) can help clarify the metabolic differences in your tumor population and help refine the number of samples needed.

## Potential pitfalls when acquiring tissue samples

To maintain the integrity of the metabolic phenotype, tissue samples are snap-frozen as soon as possible after acquisition to quench metabolic processes. This helps to preserve the labeling patterns and isotope distribution for rapid metabolic pathways, which can be depleted after prolonged periods of withdrawal from the tracer. However, two key considerations can complicate this.

First is the potential interruption of blood supply during tissue acquisition. While biopsy samples can be acquired without affecting blood flow, surgical resection can involve ligating the blood supply to the organ. A reasonable concern is how this may affect metabolite labeling within the tumor, as metabolomics studies have revealed significant differences in metabolite abundance with increased duration of ischemia. The potential extent, duration, and organ-specific effects of ischemia should be considered when acquiring these tissue samples. In some organs, residual or alternative blood supplies can sustain nutrient delivery to the tumor. For instance, when the pulmonary artery is ligated during a lobectomy, bronchial circulation can partially maintain nutrient delivery without exerting appreciable metabolic changes to the tumor (Hensley et al, 2016). The organ of interest, specific surgical procedure, expected duration of altered blood supply, and pre-existing hypoxic regions within the tumor, may each impact the observed metabolic phenotype.

A second pitfall is that even after the tumor sample is excised, it often undergoes pathological analysis before being released to research. While these delays could be partially mitigated operationally (e.g., intra-operative pathology assessment to reduce wait times before freezing), it may be helpful to test how delays in snap-freezing impact the labeling patterns in animal models. One such example examined how snap-freezing delays could affect [$^{13}$C] glucose contribution to TCA cycle metabolites. Here, enrichment in TCA cycle metabolites was not reproducibly different even after 30 minutes post-resection (Johnston et al, 2021). This suggests that while total metabolite abundances are significantly altered with prolonged ischemia, enrichment within some pathways can be reliably maintained. Establishing the retention of stable isotope enrichment (or lack thereof) in specific metabolic pathways with model systems may increase confidence in assessing the patient samples.

## Maximizing the data yield from infused tissues

Given the high costs of tracer infusions, the anticipated metabolic heterogeneity, and the overall difficulty of establishing and performing these studies, maximizing the amount of information generated from these infusions is important. Collecting patient samples in addition to the tumor, such as adjacent non-malignant tissue, blood, or urine, can help contrast the tumor with the metabolism of non-malignant tissue, measure circulating nutrient supply, and evaluate overall metabolite consumption. If clinically feasible, one informative method would be to sample both the arterial supply and venous drainage from a tumor or tumor-bearing organ, as these can provide direct evidence of the local consumption and export of metabolites from the tumor microenvironment.

The tumor sample itself can be analyzed in increasingly innovative ways, yet each approach brings both advantages and drawbacks. Most studies have analyzed bulk tissue samples, where tissues are rapidly snap-frozen. These tissues are homogenized into a single metabolite pool before analysis by mass-spectrometry, making distinguishing cell-type contributions to metabolism impossible. Several approaches have been used to separate specific cell populations, such as flow cytometry or magnetic bead isolation, enabling cell-specific measurements of metabolic features. The caveats of these approaches are that the isolation procedures (a) increase the time between sample removal and snap-freezing and (b) that the various isolation buffers may influence cell metabolism by altering metabolic gradients. Exciting advances in spatial metabolomics and mass spectrometry imaging (a collection of methodologies that analyzes metabolites in a cell-specific, spatial fashion), may offer solutions to both of these pitfalls by retaining the tissue architecture during tissue collection and metabolite analyses. While these approaches still have some technical limitations (image resolution, detection sensitivity, etc), this is a promising, rapidly evolving approach to measure isotope tracing in patient tumors.

## Interpreting labeling data from infused tissues

Our companion piece in this series offers an overview of presenting tracing data (Meiser and Frezza, 2024), as do several excellent reviews in the field (Buescher et al, 2015; Hui et al, 2020; Jang et al, 2018). Still, interpreting labeling data from patient samples can be challenging. Combined with the expected biological and tumor heterogeneity, the infused tracer can be metabolized by multiple organs, which generate other labeled nutrients that can contribute to tumor labeling patterns. While an in-depth analysis of these approaches is worthy of a dedicated review, we highlight some approaches to analyzing these data (see also Fig. 1).

Absolute enrichment and $^{13}$C label distribution is important in identifying total tracer contribution to a metabolic pathway. Comparing relative enrichment, e.g., metabolite enrichment ratios compared to the tracer (either in the circulation or the tissue itself) can be informative, especially if the research question seeks to address how much a particular nutrient contributes to a metabolic pathway. Similarly, comparing the enrichment ratio between specific labeled metabolites can approximate specific enzymatic activity. For example, the ratio of citrate $m + 2$ to pyruvate $m + 3$ can provide information related to pyruvate dehydrogenase, whereas the ratio of citrate $m + 3$ to pyruvate $m + 3$ is a labeling pattern that may be related to pyruvate carboxylase (Johnston et al, 2021). It is important to note that, while stable isotope tracing measures a nutrient's contribution to a particular pathway, it does not necessarily measure the enzymatic activity or metabolic flux. Instead, enrichment data can be used to perform metabolic flux analysis (MFA), a powerful method of inferring the relative or absolute enzymatic activity and directionality from infused samples. These methods can measure the directional movement of metabolites and pathway output (net flux), as well as metabolite exchange, cycling, and equilibrium (exchange flux). Both parameters can be used to generate a metabolic flux map, providing a comparison of metabolic pathways between tissues, and ideally, identify metabolic nodes that may be perturbed in tumors.

Relatedly, there is a growing movement to make metabolomics data publicly available (in an analogous fashion to sequencing data), and a similar approach could be taken with tracing studies. Given the difficulty and expense of performing these studies, maximizing the ability to analyze, interpret, and use these data for multiple purposes could be widely

beneficial for interested parties using publicly supported platforms like EMBL-European Bioinformatics Institute (EMBL-EBI) Metabo-Lights databases.

## Outlook: How can we utilize these data for future patient therapy?

Clinical tracing studies aim to investigate the biology of bona fide tumors and identify patient-relevant tumor biology. Stable isotope tracing can investigate processes beyond metabolite abundance and provide insights into metabolic activity within tumors. To date, these studies have provided important insights such as identifying inter- and intra-tumoural heterogeneity (Courtney et al, 2018; Hensley et al, 2016), differential substrate use by tumors, and altered metabolic pathways between tumor types (Fan et al, 2009; Ghergurovich et al, 2021; Johnston et al, 2021). In the future, long-term follow-up of infused patients may provide information on the correlation between the metabolic phenotype of a primary tumor and its metastatic potential. These approaches can be used to answer key questions, such as if metabolic phenotypes predict (or change with) resistance to tumor therapy, including chemotherapy and immunotherapy? Could specific nutrient uptake provide new insights into tumor diagnosis and patient survival? Taken together, clinical infusions have the potential to provide important insights and may have significant theranostic potential for patients.

## Key considerations

Stable isotope tracing can be performed in a clinical setting to gain unique insights into tumor metabolism. Factors such as tracer costs and inter-patient and inter-tumoral metabolic heterogeneity emphasize maximizing the information generated from these infusions.

## Peer review information

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

## Acknowledgements

The authors would like to thank Nicolas Lesner for thoughtful comments and feedback. BF is supported by the National Cancer Institute (R00CA237724) and the Cancer Research Foundation. AT was supported by an Emmy Noether Award from the German Research Foundation (DFG, 467788900) and the Ministry of Culture and Science of the State of North Rhine-Westphalia (NRW-Nachwuchsgruppenprogramm). AT was supported by the ERC Starting grant (METATARGET, 101078355). AT holds the Peter Hans Hofschneider of Molecular Medicine endowed professorship by the Stiftung Experimentelle Biomedizin.

## Author contributions

**Brandon Faubert**: Conceptualization; Funding acquisition; Writing—original draft; Writing—review and editing. **Alpaslan Tasdogan**: Conceptualization; Funding acquisition; Writing—original draft; Writing—review and editing.

## Funding

## Disclosure and competing interests statement

The authors declare no competing interests.

