## [Peer Review File · The EMBO Journal]

Approaches to stable isotope tracing and *in vivo* metabolomics in the cancer clinic

Brandon Faubert and Alpaslan Tasdogan

Corresponding authors: Alpaslan Tasdogan (Alpaslan.Tasdogan@uk-essen.de) , Brandon Faubert (bfaubert@bsd.uchicago.edu)

Review Timeline:

Submission Date:	26th Jun 24
Editorial Decision:	6th Sep 24
Revision Received:	10th Dec 24
Editorial Decision:	15th Jan 25
Revision Received:	12th Mar 25
Accepted:	17th Mar 25

Editor: Daniel Klimmeck

Transaction Report:

Dear Aslan, dear Brandon,

Thank you again for sending us your commentary article for the metabolism advice series. As mentioned, we have asked two dedicated experts to assess your manuscript, and in the meantime got feedback from both of them, which I enclose below FYI.

As you will see, the experts much appreciate the perspective piece and find it timely and worth publishing. They also provide constructive feedback on how to further improve it by advancing the discussion, and enhancing the perspectives taken e.g. broadening from analysis of bulk tumors to other sample types and amending directly metabolomics analysis-related aspects such as potential confounding effects of sample processing and other pitfalls. They also advise to make the text more concise, condensing a number of points, and to revise nomenclature.

I hope you will find the comments helpful. I am sure that an amended version incorporating the suggestions made by the referees will be highly noted and appreciated. I would thus like to invite you to submit such a revised version using the link enclosed below.

Please let me know in case I can be of any help with this.

with
Best wishes,

Daniel

Daniel Klimmeck, PhD
Senior Editor
The EMBO Journal

Referee #1:

In their commentary „Stable isotope tracing in the clinical setting" Faubert & Tasdogan aim to provide a roadmap for stable isotope tracing in cancer patients.

While the topic is relevant and interesting, the commentary failed to meet my expectations. In my opinion many of the raised points are obvious. If you want to investigate tumor metabolism you need to identify the tumor or be sure that there is one.

Tissue for metabolomics analysis should be snap-frozen immediately. Communication in the operating room is key...

Rather than describing all the downstream possibilities "histological analysis, generating patient-derived xenografts or cell lines, or collecting tumour interstitial fluid" that are options for any tissue collected, it would have made more sense had the authors focused in detail on metabolic analyses.

Histological and molecular analysis of the tumor is important to get as much information as possible from the experiments.

Regarding metabolite analysis, it is possible to analyze

- the bulk tumor tissue,
- interstitial fluid
- blood, urine, and any other available biological fluids
- sort specific cell types from the tumor tissue e.g. by flow cytometry or magnetic separation and analyse distinct cell types. However, metabolite concentrations could be altered during this procedure if cells are kept in buffers and metabolic gradients change.
- Mass spectrometry imaging could be employed for spatial analysis of metabolite distributions in the tumor tissue.

It would make sense to describe the challenges of the different approaches in detail and provide an overview of the possible analyses.

It remains unclear why generating patient-derived xenografts or cell lines would be a priority in the rare cases that metabolic flux

analyses are performed in a cancer patient, but of course if there is sufficient material, it could be interesting to compare the in vivo behavior of the tumor in the patient to that in e.g. mouse and cell models. If that was the reason to mention these possibilities, this should be clearly stated in the text.

Interesting points that would have warranted more attention are only mentioned in passing. For instance, the fact that surgery often entails interruption of the blood supply to the organ is mentioned but its consequences for metabolite tracing are not discussed.

"the deuterated hydrogen molecule", this term is incorrect. I assume the authors mean deuterium. Deuterated molecules are molecules in which the hydrogen molecules are exchanged for deuterium molecules.

Some references are missing, e.g. for: "For example, the ratio of citrate m+2 to pyruvate m+3 can provide insight into pyruvate dehydrogenase, whereas the ratio of citrate m+3 to pyruvate m+3 may indicate the activity of pyruvate carboxylase."

Taken together, while the topic of the commentary holds much promise, I would recommend to put much more work into it and to add detailed and relevant information and considerations.

Referee #2:

The manuscript titled "Stable Isotope Tracing in the Clinical Setting" provides an important and timely overview of the considerations and methodologies involved in conducting in vivo metabolic tracing studies in patients, particularly those with cancer. The authors present a comprehensive commentary on the practical challenges, technical considerations, and strategic decisions essential to start the setting up of stable isotope tracing studies in a clinical setting. Given the growing recognition of cancer metabolism's complexity and the limitations of preclinical models in fully recapitulating patient physiology, this work is highly significant. The commentary fills a gap in the literature by providing insights not only into the experimental design but also into the operational, ethical, and logistical aspects that must be addressed when translating this technique to human studies. Since the application of this technique in patients is only done in a few laboratories worldwide, this work could help more clinical researchers looking to embark on stable isotope tracing studies, offering both a high-level perspective and detailed procedural advice.

From my point of view, as a cancer metabolism researcher, it would be of great value if the following questions/points were also addressed in the commentary (if space permits):

a) Challenges and pitfalls of setting up infusions.

- Can the authors comment on how many patients could be needed to draw significant conclusions? Should patients (as tumor host) have similar characteristics or it is only needed to compare tumors of the same origin?
- Are there characteristics of the patients that need to be considered during the recruitment to ensure adequate interpretation of the data? The manuscript touches on the challenges of biological variability (e.g., differences in weight, metabolic states), but it would benefit from a more detailed discussion of how patient selection criteria (e.g., age, sex, comorbidities, prior treatments) can impact study outcomes. Given the importance of these factors in clinical studies, it is essential to guide stratification and control measures to account for variability.
- How long should infusions last? I understand it will depend on the metabolite and the pathway to be interrogated but, in this regard, some examples could be given.
- Is it needed to reach a metabolic steady state? How can this be managed?
- Can these studies be economically challenging? This could be an important limitation to highlight due to the high costs of the labeled compounds and the amount needed to reach steady-state-like.

b) Challenges and pitfalls in interpreting in vivo infusion data.

- Could the research community benefit from repositories of this type of clinical data which makes them publicly available?
- It would be important to clarify better the concept of "testing relative enrichment, e.g., metabolite enrichment compared to the tracer". Can an example illustrate what the authors want to highlight? In the same way as it is done for "ratios between metabolites" in the next sentence.
- The manuscript acknowledges the limitations of bulk tissue analysis but does not fully explore how emerging technologies, such as spatial metabolomics or single-cell tracing, could be integrated into clinical studies to address these challenges. A few sentences of discussion on how these advances can enhance data resolution and support cell-type-specific metabolic insights would strengthen the outlook section.

Minor comments to enhance readability:

- While the manuscript provides a nice overview of the technical and logistical challenges, it may benefit from a more structured approach that clearly delineates each critical step or decision point in the setup process. Currently, the text transitions between

topics (e.g., regulatory approval, tumor identification, sample handling) are less clear, which may lead to some confusion for readers who are less familiar with the field. For example, introducing subsections with distinct headers (e.g., "Regulatory and Ethical Considerations", "Operational Workflow", "Sample Processing and Handling") would enhance readability.

- While the manuscript is generally well-written, certain sections, particularly those discussing protocol design and sample handling, are somewhat dense and may benefit from additional clarity, for example, simplifying language and breaking down complex concepts into more digestible points.

- A Glossary including some perhaps less-known terms for a broader audience (such as stable isotope tracers, FDG-PET, tumor metabolic rewiring, spatial metabolomics...) could make the manuscript more accessible.

Reviewer Comments

Referee #1:

In their commentary „Stable isotope tracing in the clinical setting" Faubert & Tasdogan aim to provide a roadmap for stable isotope tracing in cancer patients. While the topic is relevant and interesting, the commentary failed to meet my expectations. In my opinion many of the raised points are obvious. If you want to investigate tumor metabolism you need to identify the tumor or be sure that there is one. Tissue for metabolomics analysis should be snap-frozen immediately. Communication in the operating room is key... Rather than describing all the downstream possibilities "histological analysis, generating patient-derived xenografts or cell lines, or collecting tumour interstitial fluid" that are options for any tissue collected, it would have made more sense had the authors focused in detail on metabolic analyses. Histological and molecular analysis of the tumor is important to get as much information as possible from the experiments.

We thank the Reviewer for this constructive feedback. We agree that some of the issues may indeed be obvious, but these were the most common problems that arose in during our efforts to teams develop these procedures. Nevertheless, the Reviewer's point is well taken, and we have shifted the tone and direction in several sections of this manuscript to address these concerns.

Regarding metabolite analysis, it is possible to analyze

- the bulk tumor tissue,
- interstitial fluid
- blood, urine, and any other available biological fluids
- sort specific cell types from the tumor tissue e.g. by flow cytometry or magnetic separation and analyse distinct cell types. However, metabolite concentrations could be altered during this procedure if cells are kept in buffers and metabolic gradients change.
- Mass spectrometry imaging could be employed for spatial analysis of metabolite distributions in the tumor tissue.

We appreciate this valuable feedback. To incorporate this, we have updated our figure and conducted a more comprehensive description of different specimens that could be collected, and provided more details on methods to analyse these samples. While an in-depth analyses of these individual approaches is beyond the scope of this manuscript, we highlight some advantages and drawbacks from these approaches in the revised text.

It would make sense to describe the challenges of the different approaches in detail and provide an overview of the possible analyses. It remains unclear why generating patient-derived xenografts or cell lines would be a priority in the rare cases that metabolic flux analyses are performed in a cancer patient, but of course if there is sufficient material, it could be interesting to compare the in vivo behavior of the tumor in the patient to that in e.g. mouse and cell models. If that was the reason to mention these possibilities, this should be clearly stated in the text.

We agree with the Reviewer that, where feasible, the generation of patient-derived xenografts or cell lines for further analysis and comparison would be a scientifically interesting avenue to pursue to facilitate a deeper understanding of the metabolic data obtained from a single patient. Given the edits for focus and clarity, the manuscript was edited to remove the section on generating patient-derived xenografts or cell lines, thus focusing solely on the patient samples and downstream analysis. Instead, we use an example of testing model systems to investigate potential pitfalls that may need to be addressed when acquiring surgically resected samples, such as time delays between resection, pathological analysis, and snap freezing.

Interesting points that would have warranted more attention are only mentioned in passing. For instance, the fact that surgery often entails interruption of the blood supply to the organ is mentioned but its consequences for metabolite tracing are not discussed.

This is a great point. To address this, we have made significant changes to the text to emphasize the importance of blood supply interruption during sampling and the potential metabolic consequences of such interventions. We also use this opportunity to describe the role of pathological analysis in resected samples, and how in many situations this must occur before tissue is released for research.

"the deuterated hydrogen molecule", this term is incorrect. I assume the authors mean deuterium. Deuterated molecules are molecules in which the hydrogen molecules are exchanged for deuterium molecules.

Thank you for catching this error. We have corrected the text.

Some references are missing, e.g. for: "For example, the ratio of citrate m+2 to pyruvate m+3 can provide insight into pyruvate dehydrogenase, whereas the ratio of citrate m+3 to pyruvate m+3 may indicate the activity of pyruvate carboxylase."

The missing reference has now been included.

Taken together, while the topic of the commentary holds much promise, I would recommend to put much more work into it and to add detailed and relevant information and considerations.

We thank Reviewer #1 for all their suggestions as it has greatly improved the quality of our work. We anticipate with these significant changes (described in above), the commentary is now suitable for EMBO Journal.

Referee #2:

The manuscript titled "Stable Isotope Tracing in the Clinical Setting" provides an important and timely overview of the considerations and methodologies involved in conducting in vivo metabolic tracing studies in patients, particularly those with cancer. The authors present a comprehensive commentary on the practical challenges, technical considerations, and strategic decisions essential to start the setting up of stable isotope tracing studies in a clinical setting. Given the growing recognition of cancer metabolism's complexity and the limitations of preclinical models in fully recapitulating patient physiology, this work is highly significant. The commentary fills a gap in the literature by providing insights not only into the experimental design but also into the operational, ethical, and logistical aspects that must be addressed when translating this technique to human studies. Since the application of this technique in patients is only done in a few laboratories worldwide, this work could help more clinical researchers looking to embark on stable isotope tracing studies, offering both a high-level perspective and detailed procedural advice.

We would like to thank the Reviewer for providing valuable, constructive feedback aimed at enhancing the overall clarity and readability of our manuscript.

From my point of view, as a cancer metabolism researcher, it would be of great value if the following questions/points were also addressed in the commentary (if space permits):

a) Challenges and pitfalls of setting up infusions.

- Can the authors comment on how many patients could be needed to draw significant conclusions? Should patients (as tumor host) have similar characteristics or it is only needed to compare tumors of the same origin?

-AND-

- Are there characteristics of the patients that need to be considered during the recruitment to ensure adequate interpretation of the data? The manuscript touches on the challenges of biological variability (e.g., differences in weight, metabolic states), but it would benefit from a more detailed discussion of how patient selection criteria (e.g., age, sex, comorbidities, prior treatments) can impact study outcomes. Given the importance of these factors in clinical studies, it is essential to guide stratification and control measures to account for variability.

The Reviewer raises very important, albeit difficult to answer, questions. Given the potential differences between cancer types and the interests of the investigator, we cannot reasonably suggest exact numbers of patients to be recruited. However, we are using the reviewers' comment to share our experience in designing these studies as feasibility and biomarkers trials, as this allows more flexibility in terms of recruitment. In the updated manuscript, we highlight the metabolic heterogeneity observed to date, both between different cancer types, and even within certain cancer types. We agree with the Reviewer that stratifying the patient population is a reasonable approach to take to control for extrinsic variables, but even so, the metabolic phenotypes of the tumors may be quite different. In the new paragraph we discuss the importance of biological variabilities and the challenges of patient recruitment and patient-level factors such as BMI, sex and age.

- How long should infusions last? I understand it will depend on the metabolite and the pathway to be interrogated but, in this regard, some examples could be given.

-AND-

- Is it needed to reach a metabolic steady state? How can this be managed?

These are excellent questions that will be of great interest to those wishing to perform these studies. Accordingly, we have updated the text to describe bolus vs. primed continuous infusions. As the reviewer points out, reaching steady state depends on a number of factors, and we have provided an overview of these for the reader.

Reaching steady-state in a shorter time can be achieved by using a priming dose. In the manuscript, we stress that while there are advantages to reaching steady state, it is not strictly necessary, as many analytical approaches are available to interrogate non-steady state tracing data. For example, the first studies in cancer patients by the Fan/Lane/Higashi group, used non-steady state, bolus doses of ^{13}C -glucose to interrogate pyruvate dehydrogenase and pyruvate carboxylase activity in lung cancer. We therefore balanced our descriptions for the potential advantages and disadvantages of each approach.

- Can these studies be economically challenging? This could be an important limitation to highlight due to the high costs of the labeled compounds and the amount needed to reach steady-state-like.

These studies can be very challenging economically, especially when considering the grade and amount of tracer that needs to be infused into patients. We have added this text when describing bolus vs. continuous infusion.

b) Challenges and pitfalls in interpreting in vivo infusion data.

- Could the research community benefit from repositories of this type of clinical data which makes them publicly available?

This is an excellent, forward-thinking question. We enthusiastically agree with this approach and hope that there will be a consensus in the field to publically deposit metabolomics and tracing data (similar to sequencing data), especially from patient studies. We have included this important aspect in our revised manuscript.

- It would be important to clarify better the concept of "testing relative enrichment, e.g., metabolite enrichment compared to the tracer". Can an example illustrate what the authors want to highlight? In the same way as it is done for "ratios between metabolites" in the next sentence.

We apologize for the ambiguity in our original statement. We have clarified the text to provide a clearer explanation of the concepts of relative enrichment between tracer and product. Similarly, we describe how ratios between metabolites can be used to infer nutrient contribution to a pool, or to identify alterations in enzyme directionality.

- The manuscript acknowledges the limitations of bulk tissue analysis but does not fully explore how emerging technologies, such as spatial metabolomics or single-cell tracing, could be integrated into clinical studies to address

these challenges. A few sentences of discussion on how these advances can enhance data resolution and support cell-type-specific metabolic insights would strengthen the outlook section.

We appreciate the Reviewer's suggestion (which was also highlighted by Reviewer 1). While an in-depth explanation for these analytical approaches is beyond the scope of this commentary, we have expanded our discussion about emerging technologies, including spatial metabolomics, in the revised manuscript. We offer a few advantages and drawbacks from these approaches, and have added references that explore these technical considerations in greater detail. We thank both Reviewers for raising this important point.

Minor comments to enhance readability:

- While the manuscript provides a nice overview of the technical and logistical challenges, it may benefit from a more structured approach that clearly delineates each critical step or decision point in the setup process. Currently, the text transitions between topics (e.g., regulatory approval, tumor identification, sample handling) are less clear, which may lead to some confusion for readers who are less familiar with the field. For example, introducing subsections with distinct headers (e.g., "Regulatory and Ethical Considerations", "Operational Workflow", "Sample Processing and Handling") would enhance readability.

We appreciate the feedback, and completely agree that our initial version could be structured in a more logical flow. The revised manuscript flows from "Study Design and Approval" to "Tracer Administration and Patient Selection", before transitioning to "Potential Pitfalls When Acquiring Tissue Samples", "Maximizing the Data Yield from Infused Tissues" and "Interpreting Labeling Data from Infused Tissues". We hope this provides a clearer guide for the reader.

- While the manuscript is generally well-written, certain sections, particularly those discussing protocol design and sample handling, are somewhat dense and may benefit from additional clarity, for example, simplifying language and breaking down complex concepts into more digestible points.

We apologize for the verbose initial version. We hope that the revised manuscript is clearer and more concise read.

- A Glossary including some perhaps less-known terms for a broader audience (such as stable isotope tracers, FDG-PET, tumor metabolic rewiring, spatial metabolomics...) could make the manuscript more accessible.

We have included a glossary for the broader audience in the revised manuscript.

Dear Aslan, dear Brandon,

Thank you for your patience with the experts' feedback.

We have now received the colleagues' additional comments. As you will see they state that the commentary has been substantially improved by your amendments, and they recommend publication by the EMBO Journal, pending minor remaining changes.

Please revisit the additional remarks by expert #1 and amend the text where appropriate. Also, I kindly ask you to complement and adjust formatting and related author information as to the list enclosed below.

I look forward to your last revision and moving ahead with acceptance and publication of this commentary shortly.

Best wishes,
Daniel

Daniel Klimmeck, PhD
Senior Editor | The EMBO Journal
d.klimmeck@embojournal.org

Instructions for preparing your revised commentary manuscript:

>> Funding: please enter 'Ministry of Culture and Science of the State of North Rhine-Westphalia (NRW-Nachwuchsgruppenprogramm)' into our online system.

>> References: correct to EMBO style, ten authors et al., and remove the DOIs.

>> Author Contributions: Please remove the author contributions information from the manuscript text. Note that CRediT has replaced the traditional author contributions section as of now because it offers a systematic machine-readable author contributions format that allows for more effective research assessment. and use the free text boxes beneath each contributing author's name to add specific details on the author's contribution.

More information is available in our guide to authors.
<https://www.embopress.org/page/journal/14602075/authorguide>

>> Keywords: please add up to five keywords for your commentary.

Referee #1:

The authors have substantially improved the manuscript in terms of its content and the clarity of the text. Regarding "Relatedly, there is a growing movement to make metabolomics data publicly available (in an analogous fashion to sequencing data), and a similar approach could be taken with tracing studies. Given the difficulty and expense of performing these studies, maximizing the ability to analyze, interpret, and use these data for multiple purposes could be widely beneficial for interested parties." This is possible using the Elixir EMBL-EBI MetaboLights database that also contains tracer studies. This is

from their homepage: "MetaboLights is a database for Metabolomics experiments and derived information. The database is cross-species, cross-technique and covers metabolite structures and their reference spectra as well as their biological roles, locations and concentrations, and experimental data from metabolic experiments. MetaboLights is the recommended Metabolomics repository for a number of leading journals." I would be great if the authors could check if MetaboLights suits the needs for the deposition of stable isotope tracing data in the cancer clinic or if there are additional requirements that are not yet met.

On another note, I was confused about the figure showing "other tissues", do the authors suggest that patients donate biopsies of other organs? I understand that collecting additional biological fluids can and should be done, however tissue distinct from that affected by cancer may not be what the authors had in mind? This should be clarified.

Referee #2:

The authors have satisfactorily addressed all reviewers' comments and provided a comprehensive response to the feedback. Revisions have significantly improved the manuscript's clarity, rigor, and scientific contribution. It will be of significant interest to the cancer research community.

The authors addressed the remaining editorial issues.

Dear Dr Tasdogan, dear Dr Faubert,

Thank you for sending us the updated final version of the commentary article.

I am pleased to inform you that your manuscript has been accepted for publication in the EMBO Journal.

Further, I will now contact our graphics illustrator to convert the commentary figure into journal style. He will contact you shortly on the proof stage image for your input.

If you have any questions, please do not hesitate to contact me.

Thank you again for your kind contribution to The EMBO Journal, which is much appreciated.

with

Best regards,

Daniel Klimmeck

Daniel Klimmeck, PhD
Senior Editor
The EMBO Journal